

# Uncertainties of SAI efficiency and impacts depending on the complexity of the aerosol microphysical model

Simone Tilmes[1], Daniele Visioni[2], Ilaria Quaglia[1], Yunqian Zhu[3,4], Charles G. Bardeen[1], Francis Vitt[1], and Pengfei Yu[5]

[1]Atmospheric Chemistry, Observations, and Modeling Laboratory, National Center for Atmospheric Research, Boulder, CO, USA
[2]Department of Earth and Atmospheric Sciences, Cornell University, Ithaca, NY, USA
[3]Cooperative Institute for Research in Environmental Sciences (CIRES), Boulder, USA.
[4]University of Colorado Boulder, USA.
[5]Institute for Environmental and Climate Research, College of Environment and Climate, Jinan University, Guangzhou, China

**Correspondence:** Simone Tilmes (tilmes@ucar.edu)

**Abstract.** Significant differences exist between Earth System Models in simulating the efficiency of stratospheric aerosol injection (SAI) experiments, particularly in terms of aerosol burden, radiative forcing, and impacts, such as tropical lower stratospheric heating and changes in ozone. However, the primary reasons for these differences have not been identified. Previous studies have proposed that these differences can be attributed to the use of different aerosol microphysical schemes,

model resolution, or other physical parameterizations. Here, we compare two sets of SAI experiments using the same modeling framework of the Community Earth System Model, differing only in their aerosol microphysical schemes: the modal aerosol model (MAM4) and the sectional aerosol model (CARMA). We analyze scenarios varying in injection location (point vs. regional), amount (5 vs. 25 TgS/yr), and material (sulfur dioxide ($SO_2$) gas vs. accumulation-mode sulfuric acid ($AM-H_2SO_4$) aerosol). Our results suggest that the SAI radiative efficiency may be substantially overestimated when using the modal aerosol

model, particularly at higher injection rates, with implications for other impacts, including stratospheric ozone. While both sets of models confirm that $AM-H_2SO_4$ injections are more effective than $SO_2$ injections in reducing net top-of-the-atmosphere radiative forcing, MAM4 yields significantly larger aerosol burdens and weaker size-dependent sedimentation, particularly at 25 TgS/yr. In contrast, CARMA produces a smaller aerosol burden, with more mass shifted into larger particles and a declining radiative efficiency at increased injection rates. These findings suggest that more sophisticated sectional models may

be necessary to accurately assess the efficacy, side effects, and climate impacts of SAI.

## 1 Introduction

Stratospheric Aerosol Intervention (SAI) has been proposed as a potential method to artificially increase the stratospheric aerosol burden and enhance the reflectivity of incoming solar radiation to space, thereby counteracting some of the impacts of global warming (Crutzen, 2006). Model simulations in recent years using comprehensive Earth System Models (ESMs) confirm

that continuous injections of sulfur dioxide into the stratosphere effectively cool the global surface temperature. However, details on the cooling efficiency per injection amount remain uncertain (Haywood and Tilmes, 2022). Multi-model comparisons



reveal differences in cooling efficiency, ranging from 0.4 to 1.3°C for 10 TgSO$_2$/yr injection (Haywood and Tilmes, 2022), or likewise, required injections range between 4 (8) and 8 (16) TgS/yr (TgSO$_2$/yr) for different models to cool 1°C, with reasons for these differences still to be understood. A multi-model comparison study based on three ESMs with interactive aerosol

microphysics further explored the differences between the injection of sulfuric acid (SO$_2$) gas and the accumulation model sulfuric acid aerosols (AM-H$_2$SO$_4$) (Weisenstein et al., 2022). The motivation for this is that SO$_2$ injections are expected to result in larger particles due to their slow condensation to AM-H$_2$SO$_4$, resulting in a radiatively less efficient aerosol size and faster removal (Pierce et al., 2010). Weisenstein et al. (2022) showed that potential injections of AM-H$_2$SO$_4$ increase the radiative efficacy (radiative forcing per unit sulfur injected), compared to SO$_2$ injections, considering injections between 5 and

25 TgS/yr. For reference, 5 TgS/yr represents annual injections of a larger volcanic eruption (e.g., Mt Pinatubo in 1991) and 25 TgS/yr represents large SAI injections that may counter the warming of a business-as-usual greenhouse gas forcing scenario by the end of the 21$^{st}$ century (Tilmes et al., 2018). They also suggested that AM-H$_2$SO$_4$ injections, rather than SO$_2$ injections, would reduce other potential side effects, including impacts on stratospheric heating, ozone, and climate.

However, considerable differences in radiative forcing between the three ESMs, which are of similar magnitude to the

differences in the injection strategies, require a more in-depth understanding of the reasons for these differences. Different models revealed significant differences in the stratospheric aerosol burden for identical injection amounts, depending on the injection material. For example, the global aerosol burden for the considered models ranges from 2.5 Tg to 6.5 Tg using SO$_2$ injections and from 4.5 to above 8 Tg for AM-H$_2$SO$_4$ injections, corresponding to 5 TgS /yr injections (Weisenstein et al., 2022). Additionally, the efficacy (radiative forcing per TgS/yr injection) of SAI varies depending on the details of the injection

location, including point injections versus regional injections, as well as the material, considering SO$_2$ gas versus AM-H$_2$SO$_4$ solid particle injections. Weisenstein et al. (2022) suggested possible reasons for the differences in SAI-induced radiative forcing responses among the three models, including model-specific differences in resolution, physics, chemistry, and QBO, as well as differences in aerosol microphysical descriptions. Concluding from their study, there are systematic agreements regarding the responses to differences in the location of the injection and material. Larger differences among the models were

found for SO$_2$ injections rather than AM-H$_2$SO$_4$, indicating that aerosol processes between the models, particularly nucleation and condensation, may differ, which affects the size distribution and therefore the effective radius. Differences also occur in the response to increasing injection rates using AM-H$_2$SO$_4$, with two models employing a modal scheme showing an increase in the aerosol burden. In contrast, the use of a sectional model shows the opposite. The resulting radiative response is dependent on the effective radius, with one modal model showing a larger effect and the other a smaller effect than the sectional aerosol

model, especially for smaller injection amounts.

Similarly, Laakso et al. (2022) compared two different microphysical models (a modal and sectional aerosol model) in the same model framework. Their paper also concludes that significant differences in SAI radiative forcing can result from the use of two microphysical models. Their paper shows that the sectional aerosol model produces substantially more radiative forcing than the modal model, due to the modal aerosol model's shortcomings in covering the entire size distribution, which creates a

gap between the two largest modes right where the scattering efficiency is maximized at 0.3 micrometers in radius (Li et al.,





2024; Murphy et al., 2020). Their study also discussed several different injection scenarios and amounts of injection; however, these scenarios differed from those in Weisenstein et al. (2022) and were therefore difficult to compare.

Here, we revisit the same experiments performed in Weisenstein et al. (2022) and solely focus on aerosol microphysical differences within the same modeling framework. Experiments are performed with the same model physics, chemistry, vertical and horizontal resolution, interactive chemistry, and a fixed QBO. One model version utilizes the Modal Aerosol Model (MAM4), and the other uses the Community Aerosol and Radiation Model for Atmospheres' sectional aerosol model (CARMA) for mixed and pure sulfate aerosol in the troposphere and stratosphere (Yu et al., 2015; Tilmes et al., 2023). In the following, we address the questions: Can the differences in the aerosol scheme alone explain most of the differences between models shown in Weisenstein et al. (2022)? What are the main reasons for these differences? What can we learn about the impacts beyond changes in radiative forcing of SAI? To answer these questions, we examine the reasons for the differences in the two model configurations regarding the efficacy of SAI, which depends on injection strategies, including regional versus point injections, as well as the material used. With a more in-depth understanding of the behavior of the two aerosol microphysical models, we discuss the impact of relevant quantities, including the magnitude of the required aerosol injection amount, stratospheric heating, and effects on ozone. The updated information will help estimate and narrow down the reasons for differences in microphysical models, as well as the consequences of a potential application based on the range of possible outcomes for cooling efficiency, climatic impacts, and potential implications for effects on acid rain and UV radiation.

## 2 Model Setup and Experiments

We use version 2.2 of the Community Earth System Model (CESM2.2), which includes configurations of the Whole Atmosphere Community Climate Model Version 6 (WACCM6) with the option of using two separate aerosol microphysical models (Tilmes et al., 2023), a modal and sectional aerosol model. The modal aerosol model (MAM4) consists of four separate internally mixed modes, with sulfate aerosol being included in three modes: Aitken, accumulation, and coarse modes (Liu et al., 2016). The Community Aerosol and Radiation Model for Atmospheres (CARMA) is used to run as a sectional aerosol model. CARMA simulates two separate groups: the pure sulfate group, which includes nucleation and condensation of sulfuric acid, and an internally mixed aerosol group that includes sulfate, black carbon, primary and secondary organic aerosol, sea salt, and dust. The model configuration was first introduced by Yu et al. (2015), coupled to CESM version 1, and then expanded to CESM Version 2 by Tilmes et al. (2023). Pure and mixed aerosol groups include self-coagulation and coagulation with the other group. The CARMA model code has been slightly updated compared to the version used in Tilmes et al. (2023), with improvements in the optical calculations when using CARMA (Quaglia et al., 2025).

The experimental setup for all model simulations utilizes the WACCM6 middle atmosphere chemistry, which features a 2-degree horizontal resolution and 70 vertical levels, with a model top at approximately 150 km (Davis et al., 2023). Quaglia et al. (2025) demonstrated that both MAM4 and CARMA accurately reproduce the observed radiative forcing and stratospheric heating from the Mt. Pinatubo eruption in 1991. Minor differences between the models reveal, in general, a better performance in using CARMA than MAM4 for larger volcanic eruptions. The model further reproduces tropospheric and stratospheric



background aerosol conditions, including aerosol optical depth, radiative forcing, heating, and their effects on ozone. This model has been further evaluated in detail in comparison to observations and both MAM4 and CARMA for background conditions (Brodowsky et al., 2024).

The experimental setup follows that described in Weisenstein et al. (2022). In particular, all experiments utilize the climatological sea surface temperature data from 1988 to 2007 of the CMIP5 PCMDI-AMIP-1.1.0 SST/Sea Ice dataset (Taylor, 2000). We further prescribed greenhouse gas concentrations, as well as aerosol and chemical emissions, for the year 2040, based on the SSP5-8.5 future scenario (O'Neill et al., 2016). A background experiment without any injections has been performed for 30 years. The injection experiments spanned 20 years, starting from the spin-up experiment after 10 years. For the analysis, we only used years 20-30, excluding the first 10 years. We performed eight injection experiments for each CARMA and MAM4, using two different materials ($SO_2$ and AM-$H_2SO_4$ particulate), two different geographical distributions of injection, and two different injection amounts. For AM-$H_2SO_4$ injections, we assumed injections into the accumulation mode for MAM4. For CARMA, we distribute the injections over corresponding bins, assuming a lognormal distribution with a dry radius of 0.1 micrometers and a standard deviation of 1.5, analogous to what has been done in Weisenstein et al. (2022) for the sectional aerosol model. The different geographical distributions included: a) point injections at 30°N and 30°S, around 21 km and at 180 E with an equal amount of injection at each of the two points, and b) regionally distributed injections (or regional injections) with equal injections throughout the region between 30°N and 30°S, 19 – 21 km and all longitudes. Two different injection amounts are 5 TgS/yr and 25 TgS/yr, distributed over each time step of the simulation. For more details, see Table 2 in Weisenstein et al. (2022).

## 3 The effects on aerosol burden and efficacy

The model experiments investigated here are expected to yield different outcomes in aerosol burden, size distribution, and effective radius, depending on the injection strategy, namely regional versus point injections and the material used (here $SO_2$ versus AM-$H_2SO_4$), as well as the amount 5 TgS/yr vs 25 TgS/yr (Weisenstein et al., 2022). We first discuss $SO_2$ injections and contrast the differences between point vs regional injections. In the second part, we discuss AM-$H_2SO_4$ injections.

### 3.1 $SO_2$ injections

$SO_2$ injected into the stratosphere by volcanoes oxidizes and forms sulfur trioxide ($SO_3$) within days to weeks. Once $SO_3$ is formed, it is quickly transformed into $H_2SO_4$ gas. $H_2SO_4$ gas then nucleates over time or condenses onto existing aerosols. For large amounts of $SO_2$ injections, e.g., after a volcanic eruption, the formation of $SO_3$ can be limited by the availability of OH, which delays the formation of sulfate (e.g., Mills et al., 2017). Since $SO_2$ point injections of a fixed amount require a larger number density of $SO_2$ to be released in one grid box than regional injections, the oxidation of $SO_2$ is expected to be more limited by OH for point injections than for regional injections. Consequently, the formation of $H_2SO_4$ gas is slowed for point injections, and as a result, nucleation into sulfate aerosol is slower compared to regional injections. In addition, $SO_2$ and sulfate aerosol are transported by the movement of the Brewer-Dobson Circulation (BDC) stratospheric circulation upwards





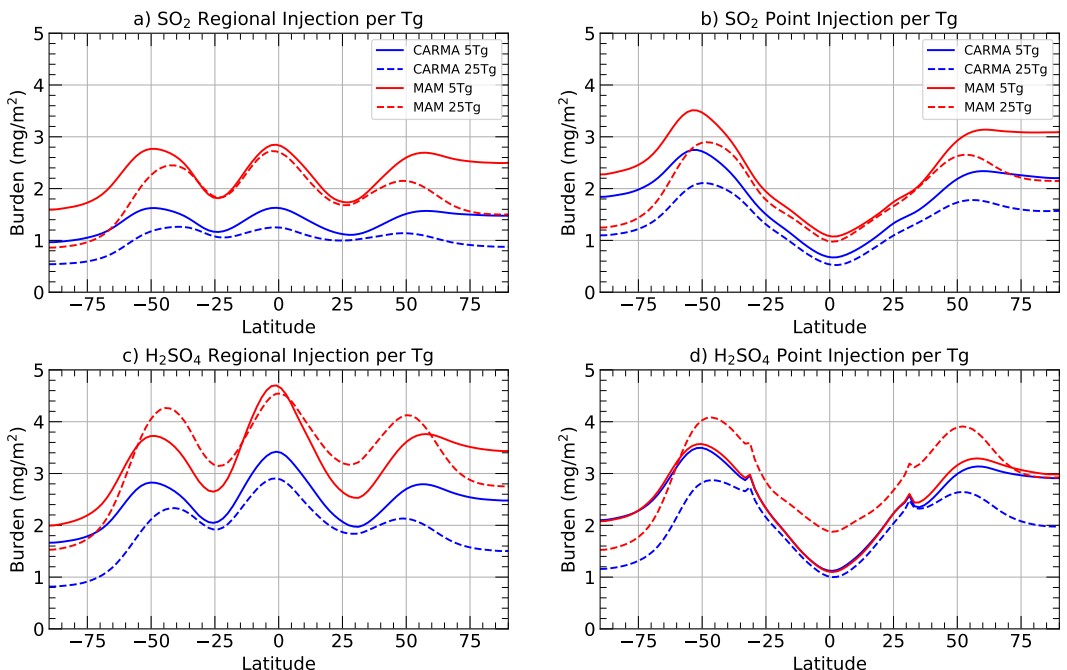

**Figure 1.** Zonal mean changes in aerosol column burden (perturbed minus background) (mg/m$^2$), scaled by the injection amount in TgS/yr, for CARMA (blue) and MAM4 (red), for different injection locations (regional injections: panels a,c, point injections: panels b,c), different injection material (SO$_2$: panels a,b, AM-H$_2$SO$_4$: panels c,d), and for different injection amounts (5 TgS/yr: solid, 25 TgS/yr: dashed).

and towards mid- and high latitudes. For point injections in 30°N and 30°S, the continuous transport direction away from the injection source enables continued nucleation and reduces the condensation or coagulation of previously formed particles. This results in a minimum sulfate burden in the tropics and a maximum in mid-to-high latitudes (Figure 1b, Figure A1).

Regional injections between 30°N and 30°S and across all longitudes are expected to result in faster nucleation at the
125 injection region, with reduced OH limitation. In addition, the continued supply of sulfur between 30°N and 30°S results in the accumulation of sulfate mass in the tropics, where the mass is somewhat constrained within the tropical pipe (Niemeier and Schmidt, 2017; Richter et al., 2017; Weisenstein et al., 2022). This leads to more substantial condensation and growth of existing particles. As for the point injections, gases and aerosols are transported toward the high latitudes through the BDC, resulting in a secondary maximum at latitudes of 40 – 60° (Figure 1a).

130 While the shape of the zonal mean stratospheric aerosol burden is similar between CARMA and MAM4 (Figure 1a,b red vs blue lines), MAM4 results in almost double the total aerosol burden per TgS/yr injection than CARMA (Figure 2a). In addition, CARMA results in a larger sulfate aerosol burden per TgS/yr injection for point injections, whereas the opposite is true for MAM4.



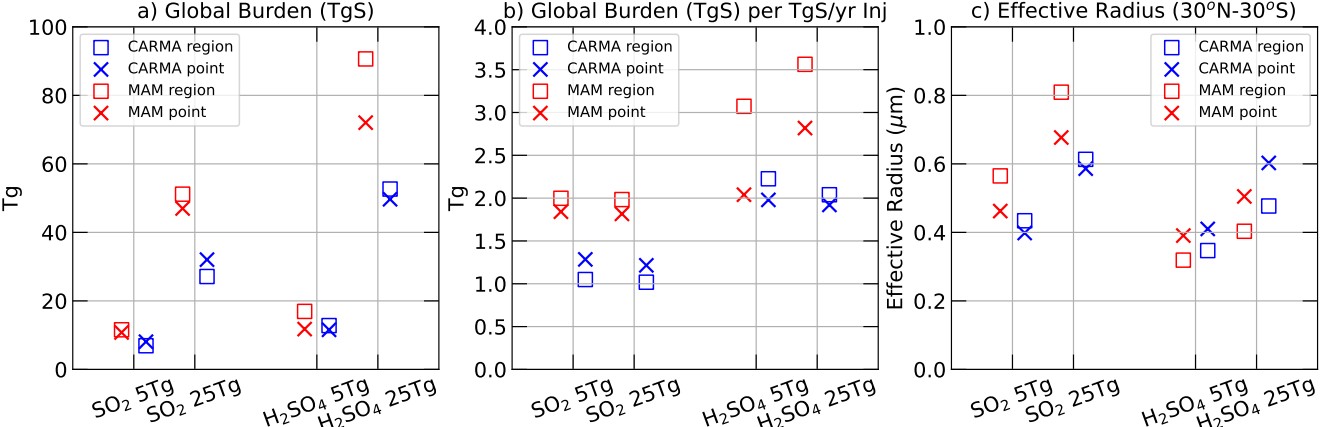

**Figure 2.** a) Global aerosol burden increase in TgS, for CARMA (blue) and MAM4 (red), for different injection locations (regional injections: squares, point injections: crosses), different injection material, and for different injection amounts as indicated on the x-axis. b) As a), but the global aerosol burden in TgS is scaled by the injection amount per TgS/yr. c) As a), but for the stratospheric effective radius (averaged between 30°N and 30°S). The effective radius is scaled with the Surface Area Density to only include regions in the stratosphere with elevated aerosol burden.

The CARMA sectional aerosol model divides the aerosol distribution into 20 bins, with the smallest bin of the pure sulfate group starting at 0.2 nm and the largest bin at 1.3 microns. In contrast, MAM4 describes the aerosol distribution using three modes: the Aitken, accumulation, and coarse modes (Figure 3, blue bars for CARMA vs red line for MAM4). The Aitken mode, with a range between 15 and 53nm, is significantly larger than the smallest CARMA bin, where the initially formed small sulfuric acid particles (sulfates) build up in the smallest bin/mode. Since the smallest nucleation size in MAM4 is significantly larger than in CARMA, sulfate molecules nucleate more rapidly, facilitating greater coagulation (see also Tilmes et al. (2023). The longer nucleation time in CARMA can further cause the recycling of $H_2SO_4$ back to $SO_2$ (through photolysis). This difference between the two aerosol models results in a slightly larger effective radius for MAM4 compared to CARMA for both point and regional injections (Figure 2c, Figure A3). A similar difference between CARMA and MAM4 has been found after the Mt. Pinatubo eruption (Tilmes et al., 2023). However, a few weeks after the eruption, the CARMA effective aerosol grew larger than in MAM4. On the other hand, both aerosol models show consistent differences between regional and point injections in the effective radius. The faster condensation and coagulation for regional injections are expected to result in larger particles in the injection regions between 30°N and 30°S, as can be seen in Figures 2c and A3).

For continuous $SO_2$ injections, the number of nucleation mode particles is larger in CARMA. In comparison, the number of accumulation mode particles is larger in MAM4 due to the faster nucleation and condensation processes that occur with MAM4 (Figure 2a, b). In addition, the volume size distribution (Figure 2c,d) shows that CARMA shifts more mass into the largest bins for both point and regional injections, resulting in stronger sedimentation. In contrast, the size range for MAM4 is restricted to smaller mass bins due to the limited sigma range of the MAM4 coarse mode, resulting in less sedimentation and a larger total

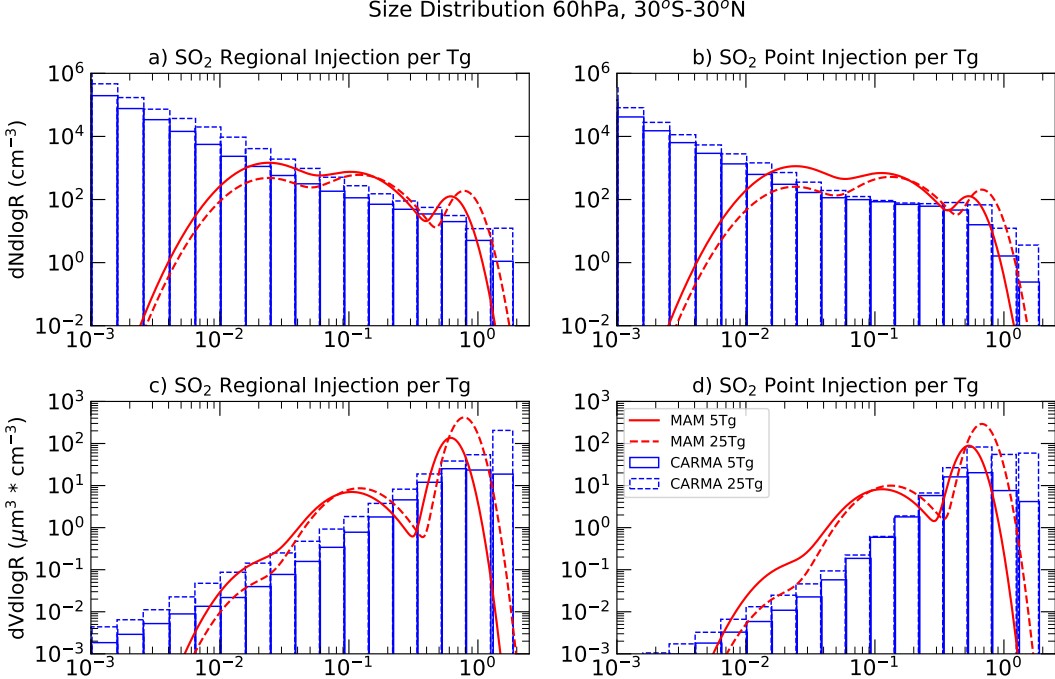

**Figure 3.** Size distribution for SO$_2$ injection experiments shown as dN/d10logR (particles cm-3, panels a,b) and dV/d10logR (cm-3 * um-1, panels c,d), at 60 hPa, averaged between 30°N and 30°S. Results are shown for CARMA (blue) and MAM4 (red), comparing different injection locations (regional injections on the left, point injections on the right), and for varying injection amounts (5 TgS/yr, solid lines; 25 TgS/yr, dashed lines).

aerosol burden. Consequently, for the same SO$_2$ injections, MAM4 exhibits a significantly larger burden than CARMA. The shift of number and mass towards the largest bin in CARMA also aligns with relatively larger sedimentation in CARMA for regional injections compared to point injections, resulting in a reduced burden for regional injections. In contrast, the larger

sizes in MAM4 align with the larger burden for regional injections compared to point injections, suggesting a relatively smaller increase in the largest mode and consequential reduced sedimentation.

For larger SO$_2$ injection rates (25 TgS/yr), continuous injections lead to a decrease in Aitken mode aerosol number density and an increase in coarse mode number density in MAM4, for both regional and point injections. In contrast, CARMA shows an increase in aerosol number density in both nucleation and Aitken modes with the larger injection rate for point injections,

and an increase across all size bins for regional injections. This is due to the continuous nucleation of particles in CARMA, whereas for MAM4, most additional sulfur condensation occurs on existing particles. This difference in behavior between the two models results in a somewhat larger increase in effective radius in MAM4, from 5 TgS to 25 TgS injections for both point and regional injections. However, both models show little reduction in burden per TgS in the Tropics, but a significant





reduction in aerosol burden in mid-to-high latitudes, likely due to the faster removal of larger aerosol particles, which is more
pronounced in MAM4.

In summary, differences in both nucleation processes (resulting in smaller particles in CARMA) and the limited coarse mode range in MAM4 are likely responsible for the differences in burden between CARMA and MAM4. For the accumulation mode, the high cutoff value in MAM4 is 0.48 microns with a sigma value of 1.6 microns. For the coarse model particles, the low cutoff value is 0.4 microns and the high value is 40 microns with a sigma value of 1.2 (Mills et al., 2016). The coarse mode's
sigma value of 1.2 seems to limit the peak of the coarse model size towards 0.5-0.6 microns for 5 TgS/yr and 0.7-0.8 for the 25 TgS/yr injections. In CARMA, sulfur injections do not result in a peak for the coarse mode number; instead, a steady decline towards larger bins is observed for most cases. Besides shortcomings in MAM4, CARMA also has shortcomings in fully resolving the sizes of the large injections. Especially for 25 TgS/yr regional injections, mass is accumulating in the largest bins, which restricts particles from growing larger and potentially alters sedimentation rates in the model.

These findings are similar to those of Laakso et al. (2022) and Weisenstein et al. (2022), who found that the sectional aerosol model favors new particle formation over condensation on existing particles. In contrast to Laakso et al. (2022), MAM4 does not create a large gap between the accumulation and coarse modes despite having a similar setup as the modal model used in their study.

## 3.2 H$_2$SO$_4$ injections

The purpose of using AM-H$_2$SO$_4$ for SAI is to keep aerosol particles to a more optimal size, which would result in more effective scattering and radiative forcing for the same mass injected (Pierce et al., 2010; Benduhn et al., 2016; Weisenstein et al., 2022). Using SO$_3$ or H$_2$SO$_4$ injections, therefore, will not require the initial nucleation of sulfate and also prevent condensational growth of injected particles. As for SO$_2$ injection, the resulting size of sulfate particles will depend on the number concentration of SO$_2$ injected per grid box. In contrast to SO$_2$ injections, point injections result in a higher concentration
of particles at the injection location, leading to more initial coagulation than regional injections (Benduhn et al., 2016). This results in a somewhat smaller effective radius for both MAM4 and CARMA compared to point injections (Figure 2c). Interestingly, point injections of SO$_2$ and AM-H$_2$SO$_4$ result in a similar effective radius for 5 TgS/yr injections. In contrast, regional injections result in a smaller effective radius for AM-H$_2$SO$_4$ than SO$_2$ injections for both MAM4 and CARMA, which aligns with the more optimal radiative range between 0.3-0.4 microns (e.g., Weisenstein et al., 2022).

As for SO$_2$ injections, regional injections initially constrain more mass in the tropics, resulting in a maximum aerosol burden (Figure 1, c,d, Figure A1). In contrast, point injections result in more mass in mid-to-high latitudes. In comparison to SO$_2$ injections, AM-H$_2$SO$_4$ injections show more than a doubling of the sulfur burden for regional injections in MAM4, which aligns with the reduced effective radius resolution and less sedimentation. In contrast, the burden of point injections is similar to that of SO$_2$ injections, with a similar effective radius.

However, larger differences between CARMA and MAM4 in aerosol burden occur for AM-H$_2$SO$_4$ injections than for SO$_2$ injections (Figure 2, a,b). The global burden per TgS injection is significantly larger for regional injections in MAM4 than in CARMA. Increased regional injections increase the efficacy (in burden per TgS/yr injection) in MAM4 for both regional and





point injections. In contrast, in CARMA, the burden per TgS/yr injection is slightly reduced. This behavior is substantially different between the two microphysical models, and can also be explained by the specifics of the bin vs mode structure.

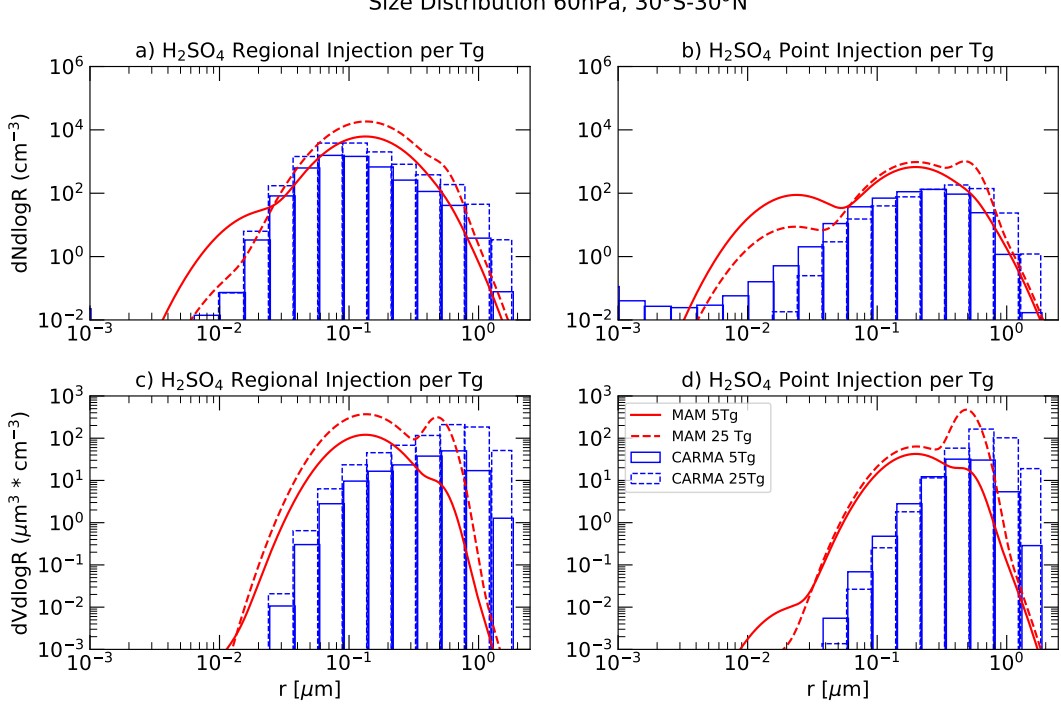

**Figure 4.** Size distribution for AM-$H_2SO_4$ injection experiments shown as dN/d10logR (particles $* cm^{-3}$; panels a,b) and dV/d10logR (cm$^{-3} * \mu$ m$^{-1}$; panels c,d), at 60 hPa, averaged between 30°N and 30°S. Results are shown for CARMA (blue) and MAM4 (red), comparing different injection locations (regional injections: panels a,c, point injections: panels b,d), and for different injection amounts (5 TgS/yr: solid lines, 25 TgS/yr: dashed lines).

As for $SO_2$, the increase of sulfur injection from 5 to 25 TgS/yr in CARMA leads to an increase in the number and mass of the largest aerosol bins, and with this, an increased sedimentation of mass (Figure 4a,b). Therefore, both regional and point injections show a substantial reduction in burden per TgS/yr injection (Figure 2a,b). For MAM4, the additional mass accumulates in the accumulation model (especially for regional injections) and in the coarse mode. However, the peak of the coarse model in MAM4 does not expand significantly towards larger sizes, unlike for $SO_2$ injections. In addition, the larger

heating of the lower tropical stratosphere in this case (discussed below) is aligned with a larger spread of sulfur towards higher altitudes. It therefore increases rather than decreases the relative burden per injection amount (Figure A2). It is further interesting that, while the sulfur burden for CARMA and MAM4 is very similar for AM-$H_2SO_4$ point injections of 5 Tg S/yr, the number and volume size distribution exhibits a very different picture. For point injections, both MAM4 and CARMA populate some Aitken model particle sizes, likely due to the evaporation of sulfate particles, which results in the additional

nucleation of smaller particles. On the other hand, the fixed sigma value MAM4 is expected to numerically push some mass





towards smaller sizes with increasing injections. This behavior is much more pronounced in MAM4, artificially reducing the effective radius.

In summary, while the importance of nucleation and condensational growth is minimized in the AM-$H_2SO_4$ experiments, the differences between the sectional and modal models result in substantially different behavior regarding changes in efficacy

with increasing injection amounts. The increase in the effective radius in both CARMA and MAM4 with increasing injection rates is similar for regional injections. However, the size distribution reveals that while both models add more mass to the coarse mode sizes, in MAM4 the peak of the coarse mode is centred around smaller sizes than in CARMA. Therefore, CARMA adds most of its mass to the largest mass bin of 1.3 microns, while MAM adds the mass at the peak of the coarse mode, which is around 0.4 microns. These differences are causing increasing sedimentation in CARMA but not in MAM4. Similar results

were also found in Weisenstein et al. (2022), where the sectional aerosol model resulted in a larger number density for larger sizes than the modal models for regional injections and a substantially larger global sulfur burden for AM-$H_2SO_4$ injections in MAM4 compared to the other models. Interestingly, in Weisenstein et al. (2022), the other modal model (with a different complexity than MAM4) resulted in a more similar or smaller aerosol burden than the sectional aerosol model.

## 4 Effects on Radiative Forcing

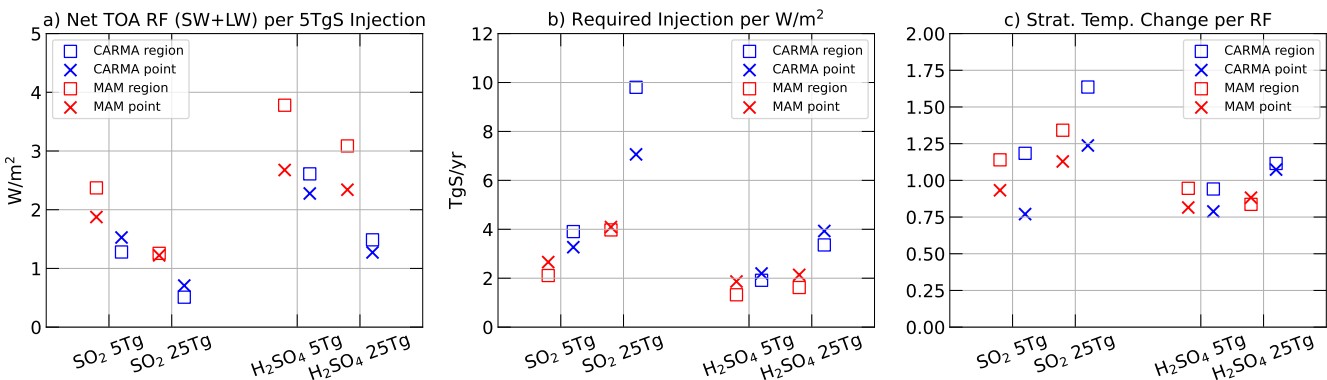

**Figure 5.** a) Globally averaged net top-of-atmosphere shortwave and longwave radiative forcing (W/m$^2$) per 5 TgS/yr injections minus the control experiments. Results are shown for CARMA (blue) and MAM4 (red), comparing different injection locations (regional injections: squares, point injections: crosses), different injection material, and different injection amounts (as indicated on the x-axis). b) As a), but for the required injection per the net top-of-atmosphere radiative forcing (W/m2). c) As a), but for temperature changes compared to the baseline simulation (averaged between 30∘N and 30∘S).

The effect of sulfur injections on radiative forcing, e.g., the reduction in the top-of-the-atmosphere (TOA) net radiative balance, defined as the sum of short-wave and long-wave flux changes, depends on the stratospheric burden and the size of the particles (e.g., effective radius). Here, we illustrate the TOA radiative forcing per 5 TgS/yr injection (instead of per 1 TgS/yr





injection, as done for the burden), for better readability of the results (Figure 5a), and refer to this as "radiative forcing (RF) efficiency" in the following.

In general, the injections of AM-$H_2SO_4$ result in a larger RF efficiency per 5 TgS/yr injections than the corresponding $SO_2$ injections (Figure 5a), in agreement with the finding from Weisenstein et al. (2022). The reason for this is that AM-$H_2SO_4$ injections result in a larger aerosol burden and a smaller (more optimal) effective radius than corresponding $SO_2$ injections. In agreement with earlier studies, increasing injections of either material (from 5 TgS/yr to 25 TgS/yr) leads to a substantial decline in the RF efficiency. We find a reduction of around 50% for CARMA and around 20% for MAM4 in AM-$H_2SO_4$

injections with increasing injection rates. While the global burden per TgS injection is very similar for the 5 and 25 TgS injection cases (especially for $SO_2$ injections), the larger injection results in a significantly larger effective radius, leading to a decline in RF efficiency. Furthermore, the reduction in efficacy in CARMA for AM-$H_2SO_4$ is significantly more pronounced because of the more substantial reduction in burden with increasing injections in CARMA compared to MAM4, as discussed above. In addition, CARMA suggests that point injections result in larger RF efficiency for $SO_2$ injections, whereas the opposite

is true for MAM4.

The difference in RF efficiency translates to the required injections needed to achieve a reduction of 1 W/m$^2$ (Figure 5b). We acknowledge that these values are based only on the 5 TgS/yr and 25 TgS/yr injection cases, which require an approximation between small and large injections. The resulting numbers differ between CARMA and MAM4 by a factor of 2.5 for large $SO_2$ injections and a factor of 2 for large AM-$H_2SO_4$ injections.

In conclusion, similar to Laakso et al. (2022) and Weisenstein et al. (2022), the details of different aerosol microphysical models can result in substantially different injection amounts to achieve the same RF, which is also the case for relatively small injections of 5 TgS/yr. For small injections, the larger RF efficiency is achieved for AM-$H_2SO_4$ injections, both point injections and regional injections, consistent with Weisenstein et al. (2022). For larger injections around 25 TgS/yr, the RF efficacy is strongly reduced for both AM-$H_2SO_4$ and $SO_2$ injections. Furthermore, the possible range of uncertainty due to the different

aerosol models, which is a factor of 5 for $SO_2$ injections (consistent with the results from Laakso et al. (2022)) and a factor of 2 for AM-$H_2SO_4$ injections, should be considered when making cost estimates of SAI injections.

## 5   Impacts on the lower tropical stratospheric temperature

The injection of sulfate aerosol into the stratosphere has been shown to result in a heating of the tropical lower stratosphere as well as changes in the aerosol surface area density (e.g., Haywood and Tilmes, 2022). These changes result in impacts

on stratospheric transport, chemistry, and ozone. In addition, stratospheric heating is an important driver of changes in the surface climate (e.g., Tilmes et al., 2025, and references therein). Temperature changes in the lower tropical stratosphere are dependent on the sulfate mass injected around the tropics (e.g., Richter et al., 2017). The larger the aerosol mass, the larger is the absorption of solar radiation and therefore the heating of the adjacent atmosphere. Here, we discuss tropical lower stratospheric temperature changes resulting from a reduction in radiative forcing for both 5 and 25 TgS/yr injection cases (Figures 5c and

A5, for 5 TgS/yr injections only).





We illustrate the tropical lower stratospheric heating changes scaled to the net top-of-the-atmosphere radiative forcing resulting from the sulfur injections (as shown in Figure 5a). We expect that the experiments with the lowest stratospheric temperature change per RF will result in the least amount of changes to the surface climate. The lowest heating per reduced RF of below 1 degree C is found for $SO_2$ point injections and $AM-H_2SO_4$ point and regional injections for both CARMA and MAM4, with

slightly smaller heating in CARMA. Regional injections generally show more heating per RF, due to the injections within the entire tropical region and a corresponding larger aerosol burden. For large $SO_2$ injections of 25 TgS/yr, experiments show an increase in heating per RF of up to 75% for CARMA and only around 20% for MAM4, resulting in more heating per RF in CARMA than in MAM4. For $AM-H_2SO_4$ injections, the heating per RF is similar for most cases, besides somewhat larger heating in CARMA for 25 TgS/yr injections.

## 6   Impact on total column ozone

The effect of SAI on total column ozone (TCO) has been discussed in detail in various studies, which point out that changes in TCO depend on injection strategies and model internal differences (Haywood and Tilmes, 2022). Most recent studies (Bednarz et al., 2023) have assessed the effects of SAI on ozone using only modal aerosol models, including MAM4. Comparisons of the effects of SAI between two microphysical models within the same modeling framework, namely CESM2, have not been

performed previously. As previously discussed (Haywood and Tilmes, 2022), the effects of SAI on ozone depend not only on chemical changes, including heterogeneous chemistry (which is proportional to changes in surface area density and also depends on temperature changes), but also on dynamical responses and transport, resulting in regions with both increasing and decreasing ozone concentrations (Figure A6).

As discussed above, point injections in 30°N and 30°S are expected to lead to stronger movement of aerosol mass towards

high latitudes and therefore result in a higher increase in Surface Area Density. In contrast, regional injections result in more mass being refined in the tropics, leading to relatively smaller SADs in high latitudes, as shown in Figure A7. On the other hand, regional injections produce stronger heating in the lower tropical stratosphere, which, in turn, affects the upwelling and transport of aerosols. They also strengthen the polar vortex, resulting in cooler temperatures in the polar vortex, due to the stronger horizontal temperature gradient in the lower stratosphere compared to point injections (e.g., Tilmes et al., 2017). The

combined changes result in a relative reduction in annual TCO in mid- and high-latitude regions, and some minor changes in the tropics (Figure 6). Interestingly, the combined changes in surface area density, temperature, and transport result in very similar changes between regional and point injections, with around 10% reductions in the SH polar region and 0-5% reductions in annual TCO in the NH polar region for $SO_2$ injections for 5 TgS/yr for both CARMA and MAM4. Stronger $SO_2$ injections of 25 TgS/yr result in larger changes, with around 16% and 18% reductions for CARMA compared to regional and point

injections, respectively, and 22% and 25% reductions in MAM4, due to the larger aerosol burden in MAM4 compared to CARMA. Strong injections can also lead to up to 10% reductions in TCO, with the most considerable changes for regional injections using $AM-H_2SO_4$.

 

For AM-H$_2$SO$_4$ 5TgS/yr injections, MAM4 is showing very similar results between regional and point injections in high latitudes, around 15% for SH and 3-6% for NH high latitudes of annual reductions. While much more mass and SAD are located in the tropics for regional injections, SAD is more similar in the high latitudes (Figure A7). Larger differences are simulated using CARMA, with 7% and 12% changes for regional injections compared to point injections, respectively, in the SH polar region, and around 3-5% in the NH polar region. Differences between MAM4 and CARMA in TCO for regional injections of AM-H$_2$SO$_4$ are aligned with the much smaller aerosol burden for regional injections in CARMA than in MAM4, discussed above. For AM-H$_2$SO$_4$ 25TgS/yr, the differences between CARMA and MAM4 are becoming more pronounced, with over 30% reductions in the SH polar region for MAM4 and over 20% for CARMA, and over 20% reduction in the NH polar region for MAM4 and around 10% reduction for CARMA. Differences are again the result of a substantially larger aerosol burden resulting from the same injection amount in MAM4 vs CARMA.

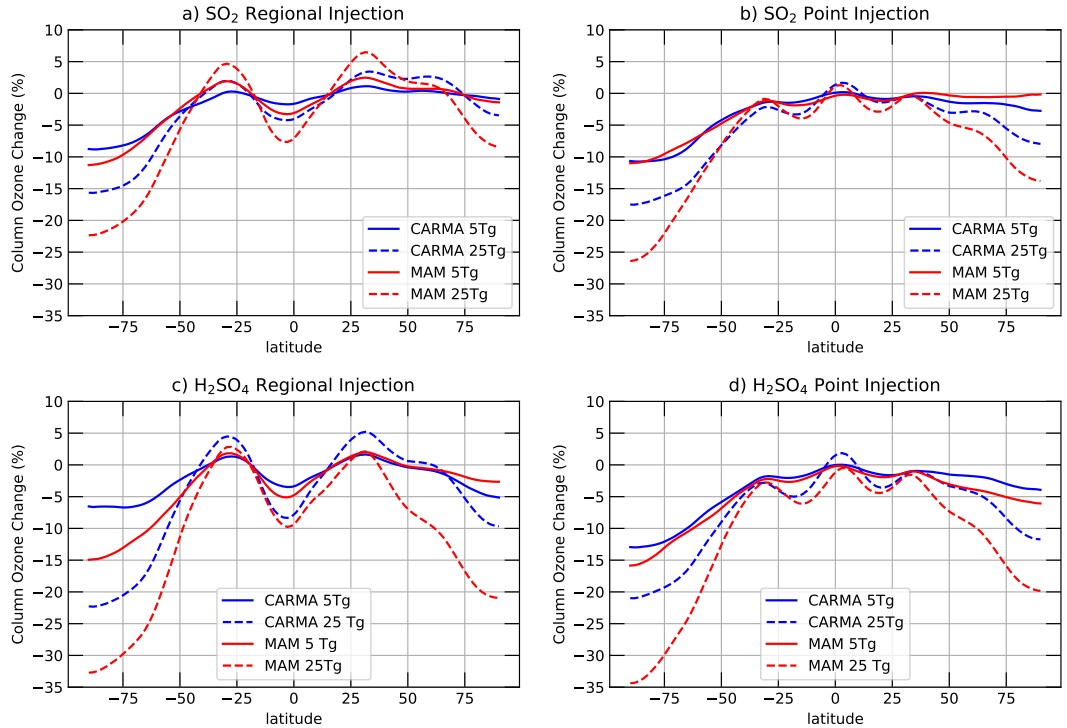

**Figure 6.** Annual relative changes in total column ozone (TCO) compared to the control for CARMA (blue) and MAM4 (red), and for different injection locations (regional injections: panels a,c, point injections: panels b,c), different injection material (SO$_2$: panels a,b, AM-H$_2$SO$_4$: panels c,d), and for different injection amounts (5 TgS/yr: solid, 25 TgS/yr: dashed), assuming 2040 halogen and greenhouse gas emissions.

In addition to comparing TCO changes directly, we also compare TCO changes scaled by the net top-of-the-atmosphere radiative forcing resulting from the sulfur injections (as also shown for temperature changes above). Since the RF efficiency is smaller in CARMA than in MAM4, the relative changes in TCO per RF result in about 2% stronger ozone reduction of





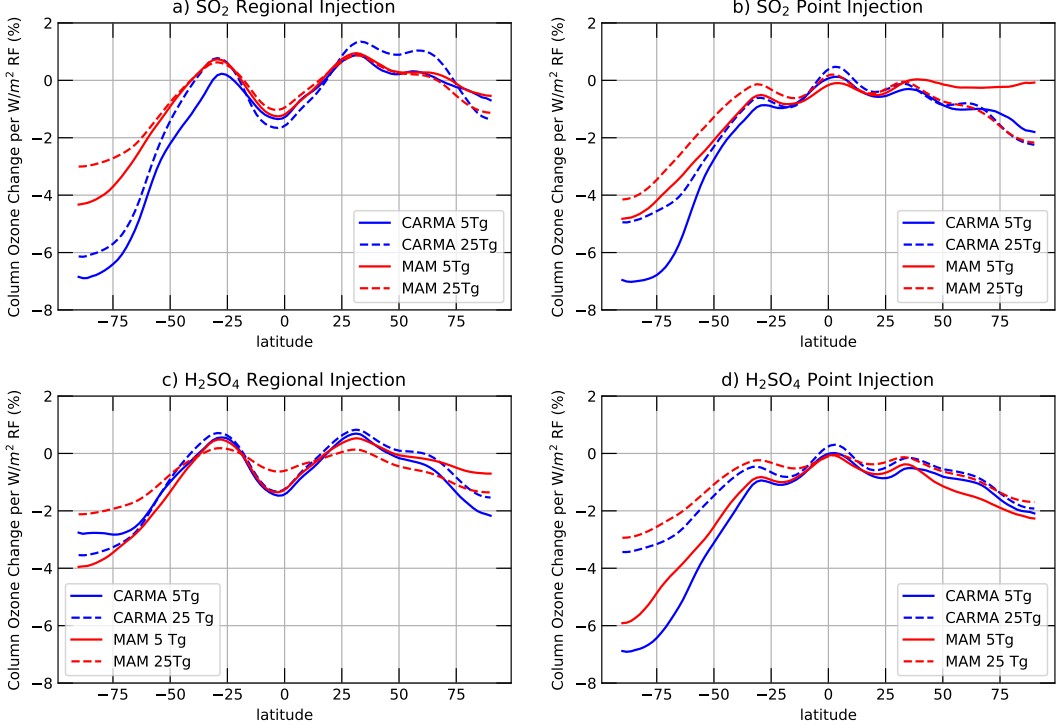

**Figure 7.** Annual relative changes in total column ozone (TCO) compared to the control divided by the net top-of-the-atmosphere radiative forcing resulting from the sulfur injections for CARMA (blue) and MAM4 (red), and for different injection locations (regional injections: panels a,c, point injections: panels b,c), different injection material (SO$_2$: panels a,b, AM-H$_2$SO$_4$: panels c,d), and for different injection amounts (5 TgS/yr: solid, 25 TgS/yr: dashed), assuming 2040 halogen and greenhouse gas emissions.

TCO per W/m2 RF in CARMA than in MAM4, especially for NH high polar latitudes and for SO$_2$ injections (Figure 7). In contrast, AM-H$_2$SO$_4$ injections show fewer differences between CARMA and MAM4. In general, larger injections of 25 TgS/yr compared to 5 TgS/yr result in relatively smaller ozone loss per RF. This is because a larger mass and effective radius result in a smaller SAD. More substantial TCO reduction per net TOA RF for the sectional aerosol model compared to the modal aerosol model was also reported in Weisenstein et al. (2022), which is also aligned with a more substantial Net TOA RF reduction in CESM compared to the sectional aerosol model in their study.

## 7 Discussion and Conclusions

The comparison of experiments using two different aerosol microphysical models within the same CESM2 modeling framework reveals that differences in the aerosol scheme alone can lead to significant variations in the effects of SAI. Therefore, other inter-model differences, as discussed in Weisenstein et al. (2022), are of reduced importance in our study. Here, we demonstrate that differences in the nucleation process can lead to significant differences in the aerosol size distribution between the





sectional and modal aerosol models. Continuous $SO_2$ injections result in more nucleation in CARMA than MAM4, resulting in a smaller effective radius in CARMA compared to MAM4, similar to what has also been found in Laakso et al. (2022). An improved nucleation scheme in MAM4 may therefore improve some of the differences between the two aerosol models.

Furthermore, the details of the mode structure in MAM4 do not reproduce the same size distribution of continuous aerosol injections as shown in CARMA. The coarse model features a relatively small sigma of 1.2 microns, which limits the growth of coarse-mode particles in MAM4 compared to CARMA. In contrast, CARMA suggests an accumulation of mass in the larger bins with increasing injection amount. The difference in coarse-mode number between CARMA and MAM4 results in a stronger removal of mass in CARMA, and therefore, an aerosol burden that is, in part, almost half the amount of that for

MAM4, especially for point injections. Therefore, changes in the coarse mode quantities in MAM4 need to be further explored to achieve a more comparable representation to CARMA and other aerosol models. For AM-$H_2SO_4$ injection experiments, nucleation and condensation are less important. Nevertheless, differences as a result of the bin structure are becoming more apparent for larger injections. MAM4 results in an increase in efficacy with increasing injections, while CARMA shows a reduction.

It has been shown that CARMA is more effective in reproducing size distributions compared to observations after the large volcanic eruption of Mt. Pinatubo in 1991. While a large, impulsive volcanic eruption might not be a suitable analogue for the continuous sulfur injections required for SAI applications, the comparisons provide us with more confidence in CARMA results than in MAM4 results. However, even CARMA may not have large enough size bins, especially to simulate the large sulfur particles resulting from SAI for very large injection cases of 25 TgS/yr, which are sufficient to counter a business-as-

usual warming scenario by the end of the 21st century (Tilmes et al., 2018). This shortcoming is likely also a potential issue in other existing aerosol models, requiring more work.

We further investigate trade-offs between different injection scenarios in terms of radiative efficiency, the required injection amounts needed to reduce the net TOA imbalance to 1 W/m$^2$, as well as the tropical lower stratosphere heating and total column ozone loss per RF. Both MAM4 and CARMA show improved RF efficiency for AM-$H_2SO_4$ injections compared to

$SO_2$ injections, in agreement with Weisenstein et al. (2022). However, CARMA RF efficiency is substantially lower by up to half the amount, and even more for 25 TgS/yr injections of $SO_2$. This difference translates into a difference in the required injections for achieving a negative RF of 1 W/m$^2$, a factor of 5 for $SO_2$ injections, and a factor of 2 for AM-$H_2SO_4$ injections for large injection amounts. The differences in RF efficiency are a result of differences in the aerosol size distribution (effective radius), as well as the aerosol burden per injection amount. This leads to similar RF efficiency only for point injections of AM-

$H_2SO_4$ between CARMA and MAM4, and larger RF efficiency in MAM4, primarily due to the significantly larger burden in MAM4 and also due to a somewhat smaller effective radius for AM-$H_2SO_4$ injections. Similar values were reported in Laakso et al. (2022), comparing a sectional and modal aerosol model. In Weisenstein et al. (2022), CESM2 MAM4 is showing the strongest RF efficiency, while the other models are more similar to the CARMA RF efficiency.

Estimations of costs need to carefully consider the range of uncertainties regarding the injection amount and the substantial

decline in efficacy with increasing injections, which is only partially taken into account (Smith, 2020). These studies also don't



consider possible differences in the injection strategy (location and material), which also have consequences for the injection amount and, therefore, the resulting costs.

Differences in the aerosol burden per TgS/yr injections and the RF efficiency between CARMA and MAM4 also translate to differences in the tropical lower stratospheric heating per unit of RF. While CARMA shows somewhat smaller or similar heating per unit of RF for point injections for 5 TgS/yr injections of both $SO_2$ and AM-$H_2SO_4$, substantially larger heating per unit of RF is simulated for CARMA than MAM4 for larger injections of 25 TgS/yr. Also, regional injection around the tropics results in stronger heating per unit of RF, due to the accumulation of more mass in that region. The smallest heating per unit of RF is simulated with large regional AM-$H_2SO_4$ injections using MAM4, which is a result of the increasing efficacy of MAM4 with increasing injections. Differences in CARMA compared to MAM4 demonstrate that many recent SAI model simulations performed with MAM4 point injections (e.g., Tilmes et al., 2018, 2020; Richter et al., 2022) may overestimate the tropical stratospheric aerosol heating response per unit of RF for smaller injections and underestimate it for large injections. The difference in the results highlights a significant uncertainty in the SAI response to surface climate when using different aerosol models.

Finally, the effects of SAI on TCO differ depending on the injection region, location, and amount of injection. However, comparing the TCO change with regard to the net top-of-the-atmosphere radiative forcing reduction per experiment reveals smaller differences between the various injection strategies, with point injections at 30°N and 30°S showing slightly larger TCO reductions in high latitudes for 5 Tg S/yr injections. Differences between CARMA and MAM4 are considerable, with CARMA generally showing larger TCO ozone in high latitudes. Significantly more ozone loss occurs in the tropics due to regional injections, resulting in larger changes in heating and transport.

In summary, we have demonstrated that considering different aerosol microphysical models can lead to significantly different impacts of SAI, including injection requirements for achieving a specific negative RF, with consequences for costs, heating of the lower tropical stratosphere, and effects on ozone. Furthermore, while AM-$H_2SO_4$ has a stronger efficacy and RF efficiency resulting in reductions in implementation costs (with the caveat that a uniformly distributed injection of AM-$H_2SO_4$ in the size of a CESM gridbox may not be technically feasible (e.g., Pierce et al., 2010), the benefits of climate impacts or the impacts on ozone with regard to the RF efficiency are often smaller compared to differences between aerosol microphysical models. The differences between CARMA and MAM4 further increase with increasing injection amounts. More work is therefore required to evaluate and improve aerosol microphysical models, and the need for using more sophisticated sectional aerosol models for simulating SAI injections is evident.

*Code availability.* The code of the model used in this work is available through Zendo (https://doi.org/10.5281/zenodo.16948605, Vitt and the CESM CARMA Development Team, 2025), and GitHub (git clone -b carma-trop-strat16, http://github.com/fvitt/CAM.git, last access: August 25th, 2025).



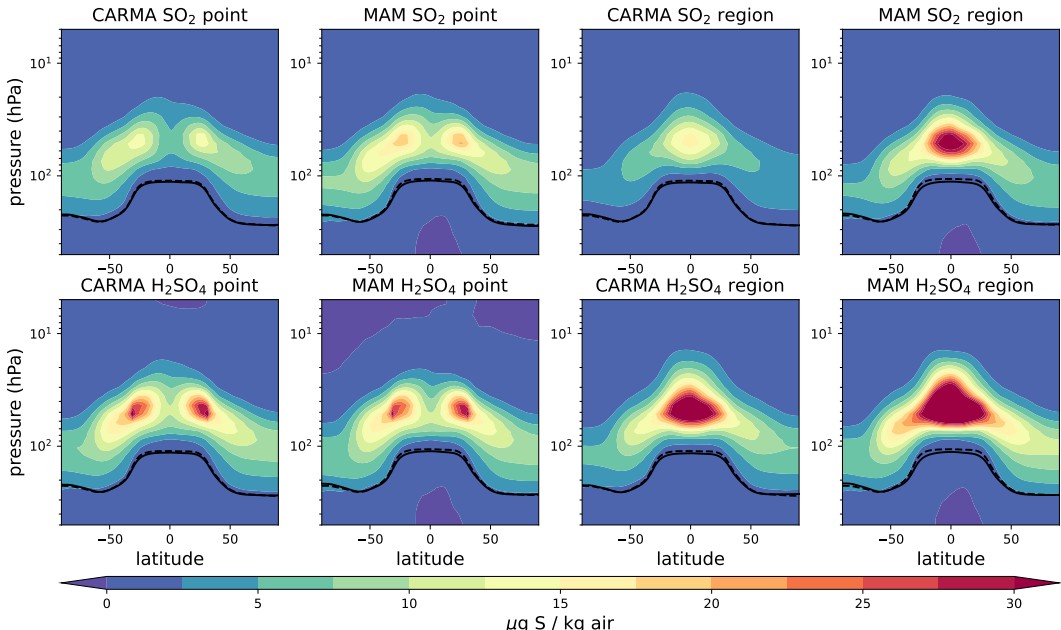

**Figure A1.** Zonal mean sulfate aerosol mixing ratios for different 5 TgS injection experiments minus the control in $\mu$g S/kg air. The tropopause of the control experiment and the injection experiments are illustrated as black solid and dashed lines, respectively.





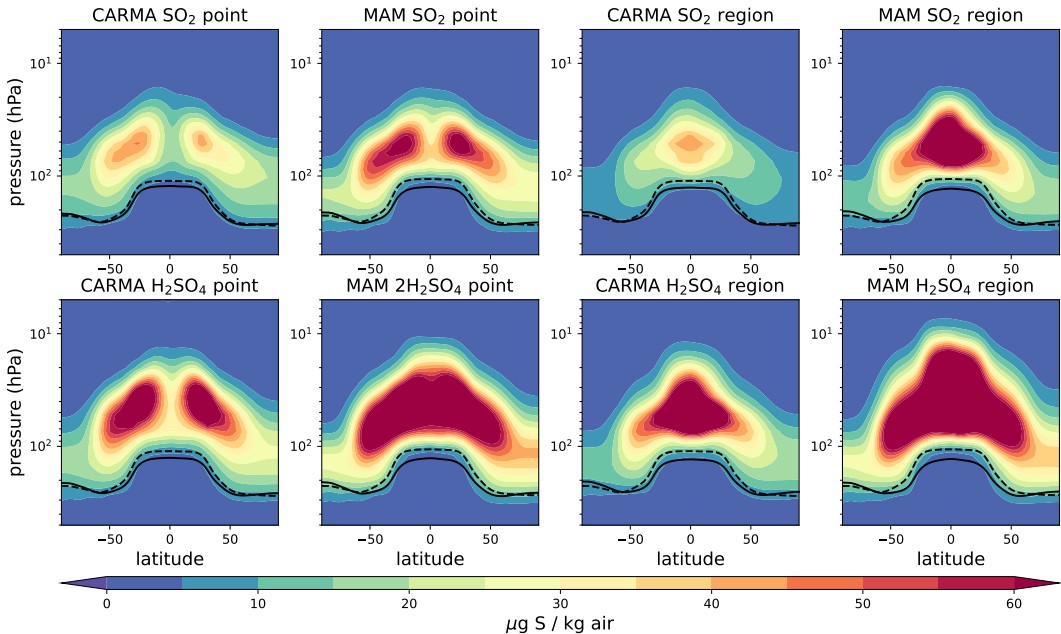

**Figure A2.** Zonal mean sulfate aerosol mixing ratios for different 25 TgS injection experiments minus the control in $\mu$g S/kg air. The tropopause of the control experiment and the injection experiments are illustrated as black solid and dashed lines, respectively.

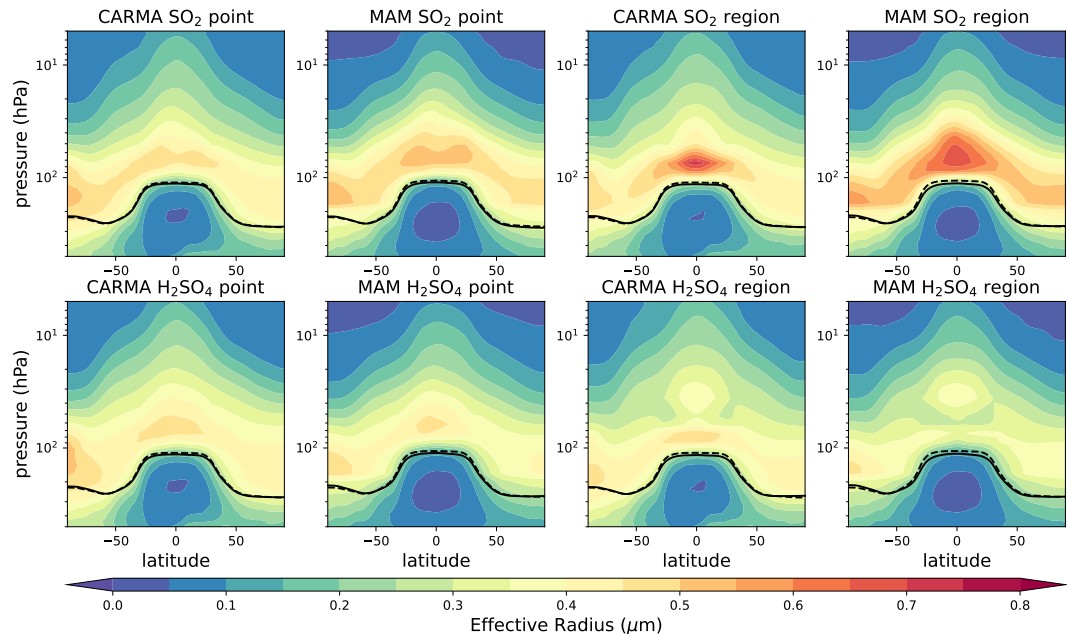

**Figure A3.** Zonal mean effective radius distribution in $\mu$m S/kg for different 5 TgS injection experiment. The tropopause of the control experiment and the injection experiments are illustrated as black solid and dashed lines, respectively.



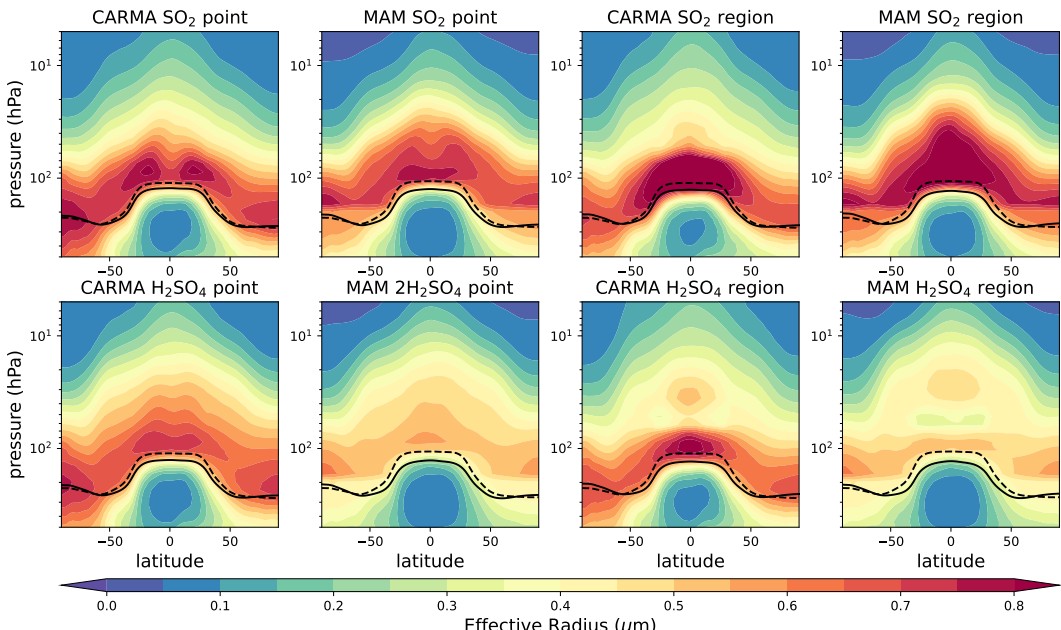

**Figure A4.** Zonal mean effective radius distribution in $\mu$m S/kg for different 25 TgS injection experiment. The tropopause of the control experiment and the injection experiments are illustrated as black solid and dashed lines, respectively.

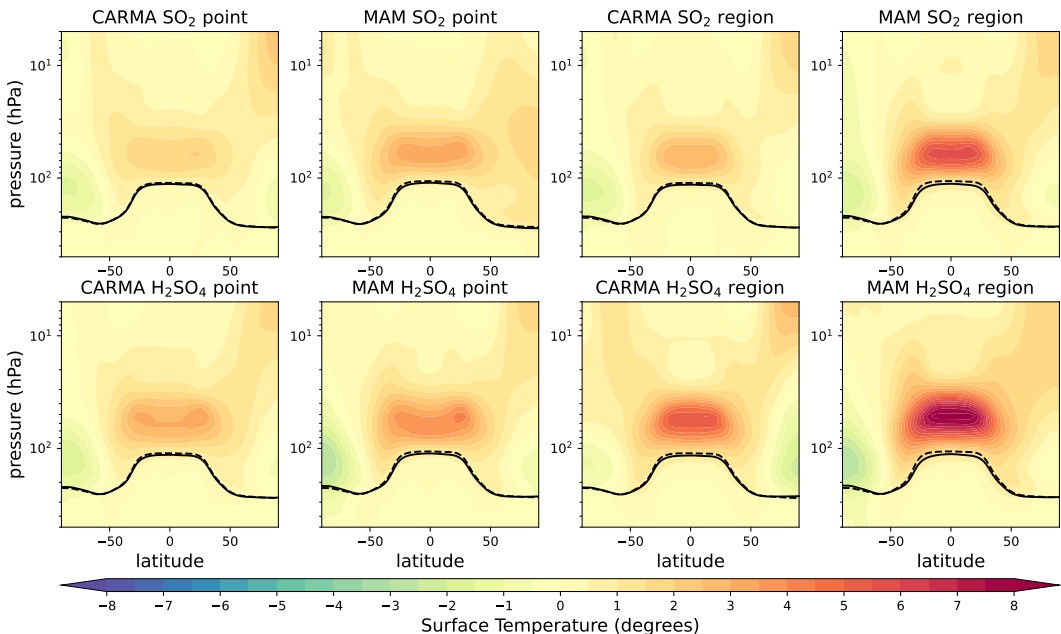

**Figure A5.** Zonal mean temperature differences for different 5 TgS injection experiments minus the control in degrees Celsius. The tropopause of the control experiment and the injection experiments are illustrated as black solid and dashed lines, respectively.



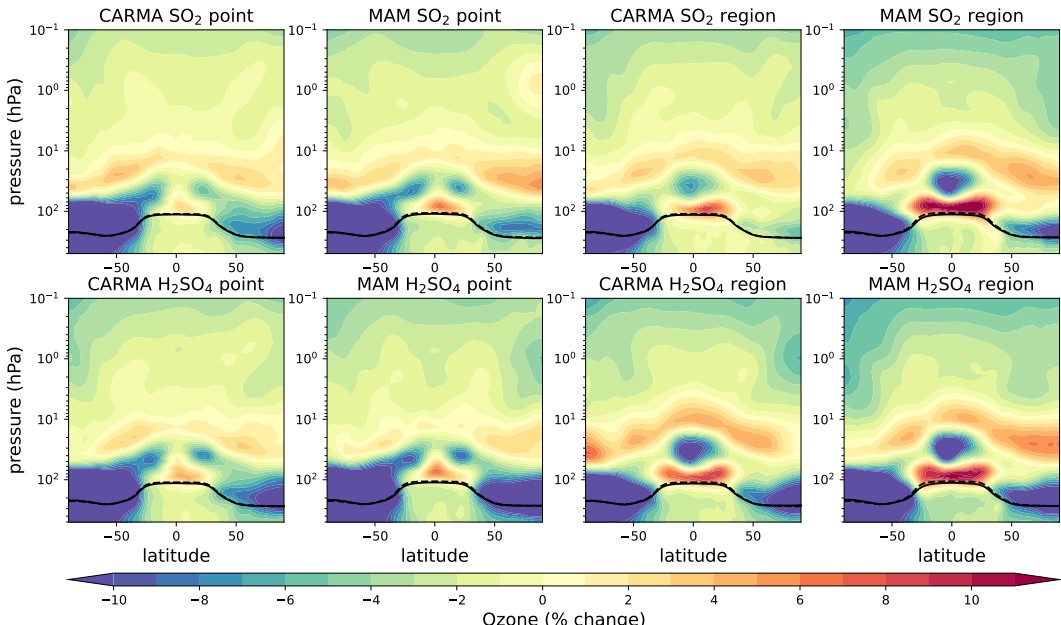

**Figure A6.** Zonal mean ozone mixing ratio relative differences for different 5 TgS injection experiments minus the control in percent. The tropopause of the control experiment and the injection experiments are illustrated as black solid and dashed lines, respectively.

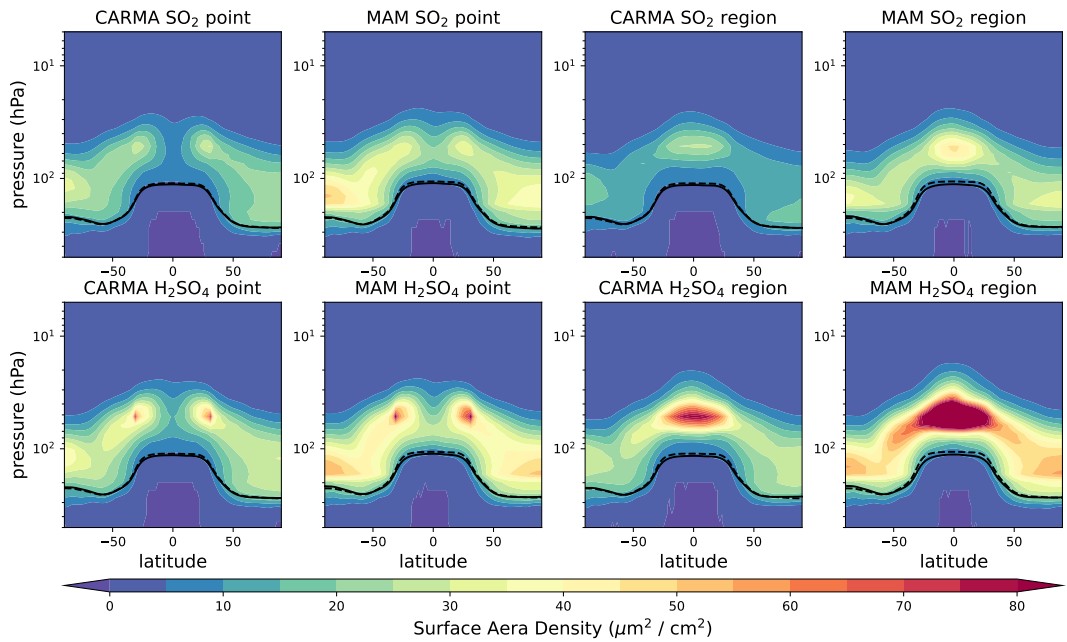

**Figure A7.** Zonal mean surface area density differences for different 5 TgS injection experiments minus the control in $\mu m^2/cm^2$. The tropopause of the control experiment and the injection experiments are illustrated as black solid and dashed lines, respectively.



*Author contributions.* S.T. outlined and wrote the manuscript. S.T. and D.V. performed model simulations. S.T. and I.Q. analyzed the results. Y.Z., C.B., and P.Y. contributed to the model development, and all authors participated in the discussion of results and reviewed the manuscript text.

385 *Competing interests.* S.T. is a member of the editorial board of ACP.

*Acknowledgements.* We acknowledge Mike J. Mills for helpful comments and discussions. The CESM project is supported primarily by the National Science Foundation. This material is based upon work supported by the National Center for Atmospheric Research, which is a major facility sponsored by the NSF under Cooperative Agreement No. 1852977. Computing and data storage resources, including the Cheyenne supercomputer (doi:10.5065/D6RX99HX), were provided by the Computational and Information Systems Laboratory (CISL) at

390 NCAR. S.T. acknowledges support from the NOAA Climate Program Office Earth's Radiation Budget Awards Number 03-01-07-001 and NA22OAR4310477. I.Q. acknowledges support from the US Simons Foundation (grant ref. MPS-SRM-00005203). Y.Z. acknowledges the support by the National Oceanic and Atmospheric Administration (grant nos. 03- 01-07-001, NA17OAR4320101, and NA22OAR4320151).



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
