# Peer review of "Uncertainties of SAI efficiency and impacts depending on the complexity of the aerosol microphysical model"

_EGUsphere, 2025_

## Author Comment (AC1)

Response to Reviewers

We thank both reviewers for their helpful comments and suggestions, which we address in detail below. As a general remark upfront, we updated three figures in the revised version of the manuscript: Figure 2, panels a and b, and Figures 3 and 4. In Figure 2, we plotted the global burden in $TgSO_2$, but the title and figure caption showed $TgS$. Therefore, we correct the figure and now actually plot the burden in $TgS$. These changes do not affect the results or conclusions in the text. For Figures 3 and 4, we corrected a mistake in our plotting script for the modal aerosol model size distribution. The MAM4 size distribution, especially for H2SO4 injections (Figure 4), aligns better with the CARMA model. However, the same shortcomings in the nucleation mode for $SO_2$ injections and in the coarse mode in MAM4 persist. Finally, we changed the caption in Figure A5 from "Surface Temperature" to "Temperature" and added a zonal mean comparison of water vapor to Appendix A8, following the comment by Reviewer 1.

**Reviewer 2**

There's some very good work in this paper. There have been other studies looking at modal vs sectional microphysical models, and the models show different answers, so it's worth seeing what another model shows.

Most of my general comments stem from the fact that I was hoping to see direct comparison with other models, like the simulations from Laakso et al. It's very important to examine how different models differ in their aerosol microphysical uncertainties.

Yes, we agree that the comparisons with other studies are important. The paper includes a bit of a discussion on this already, and in the introduction, we outline the different studies that looked at similar experiments. We are comparing our results directly to those from Weisenstein. We cannot directly compare our results to the study by Laakso et al. (2022) since the experiments were not identical; however, we are comparing the conclusions of those studies with our results. To address the reviewers' comments, we adjusted this sentence:

Introduction:

Their [Laakso et al.,] study also discussed several different injection scenarios and amounts of injection; however, **since their scenarios differed substantially from those in Weisenstein et al. (2022) and our study, we cannot directly compare the results, but discuss similarities and general conclusions**.

The questions answered in the present work seem to be somewhat niche by comparison. I like the last science question posed in Section 1 (lines 64-65), but I don't really feel like it was answered.

We agree that we only briefly touched on the last question (What can we learn about the impacts beyond changes in radiative forcing of SAI?) and only included a few impacts beyond radiation, including lower tropical stratospheric heating and effects on ozone. This certainly needs to be considered in more detail in future studies. Here, since the study is rather idealized and uses fixed SSTs, it makes most sense to focus on atmospheric changes, such as stratospheric transport and ozone. To address the comment, we add in the conclusions: "**The difference in the results highlights a significant uncertainty in the SAI response to surface climate, not only across different injection strategies but also across different aerosol models. To**

**investigate the full effects of these differences on climate, future studies using an Earth System model with a fully coupled ocean are required. "**

For instance, I struggled with the novelty of looking at point vs regional injections (see, for example, English et al., 2012 or Niemeier et al., 2013). Figure 1 illustrates this point quite well, in that the aerosol burden increases where the aerosols are injected for both MAM and CARMA, which one could have hypothesized prior to doing any simulations. CARMA shows a systematic low bias (or MAM shows a systematic high bias – it's hard to say which), but you didn't need to spread out the injection to learn that. I think the purpose of looking at this particular aspect needs to be better justified, especially with regard to what fundamental uncertainties this study is aiming to solve.

The reviewer is correct that the experimental design is not novel (which we don't claim), however, the conclusions from comparing two aerosol models are. We have repeated the same experiment as shown in Weisenstein et al. (2022) to address whether differences between the models are due to aerosol microphysics or other factors, including model resolution, physics, and transport representations. Here, we show that significant differences, similar to the inter-model spread reported in Weisenstein et al. (2022), can arise within the same Earth System model due to the use of different aerosol microphysical schemes. We think that our comparison of all the experiments shown in Weisenstein et al. (2022) is valuable, since it identified main differences between the two microphysical models, e.g., the burden per injection for CARMA is lower for regional injections and higher for MAM4 for point injections, which is of high relevance considering the injection strategies usually used in model simulations of CESM2(WACCM6) using MAM4. This shows that using different microphysicals can greatly affect the outcomes of different injection strategies. Furthermore, the Weisenstein et al. (2022) protocol was proposed as a GeoMIP testbed experiment, which we followed here. Without all the experiments, we could not have provided a full analysis as we did.

Studying the microphysics of accumulation mode aerosols are fine from a purely scientific standpoint, although given the highly questionable feasibility of this method, I would hope for a better tie-in. That is, how can we use accumulation mode injection to learn about microphysics and stratospheric processes more generally? (See my point above about the science questions.) If the only justification is that it's a proposed type of aerosol, then I put it on par with studying diamond aerosol injection – interesting but ultimately just a modeling exercise. Lines 180-189 provide a good illustration of what I'm talking about. These lines basically said that some people proposed an idea, and that's it. You could have rephrased this to be more scientifically interesting – in the previous section, you found that the nucleation stage is critical, and AM-H2SO4 allows you to isolate nucleation from coagulation growth, allowing you to further narrow which processes contribute to uncertainties.

We agree, from the process level understanding, accumulation model injections are indeed helpful to remove the effects of nucleation differences, as already discussed in the text. The use of AM particle injections has been motivated in Weisenstein et al. (2022) and other previous studies. Here, we have repeated the study to reproduce it with a different aerosol microphysical model.

All of that said, the study does add to the knowledge base in general, and the study is done well. I don't see any faults in the analysis.

Specific comments:

Line 23: I don't understand what this means. Just write it out please.

We rewrote the sentence to: "Multi-model comparisons reveal differences in cooling efficiency, ranging from 0.4 to 1.3∘C for 10 TgSO2/yr injection (Haywood and Tilmes, 2022), or likewise, **the required injections to reduce global surface temperature by 1∘C, range between 8 and 16 TgSO2/yr, with reasons for these differences still to be understood.**

Line 60: Why did you use a fixed QBO?

The model version with 2-degree horizontal resolution and 70 vertical levels cannot resolve a realistic QBO (also discussed in Davis et al., 2022).

Lines 92ff: The experimental setup seems strange. Why do you need 30-year simulations with fixed SSTs? I suppose there's nothing wrong with doing extra, but it seems like overkill.

We require a 10-year spin-up of the baseline simulation because the model has to adjust to the specific conditions with GHGs adjusted to the year 2040 and SSTs fixed at the present day. Furthermore, a robust change in ozone requires a longer period to be assessed.

Lines 147-156: Is there anything particularly surprising in these results? Surely this parallels results that others have found.

Yes, this is indeed surprising, and parallel results have not been found in other studies. We add a sentence to clarify the importance of the result: "**These findings show that the specifics of aerosol microphysical schemes can lead to opposing conclusions about whether point or regional injections result in larger sulfate aerosol burdens, which could influence decision-making about which injection scenario is preferred**."

Lines 175-178: I wanted to see more about this. This is the really interesting stuff.

We think that this part really supports the fact that sectional aerosol models behave similarly to each other when it comes to new particle formation.

Line 190: Well, yes, because you put more injection in the tropics, so of course you're going to see more aerosol there.

The point of this part is that the size of aerosol particles strongly depends on the number density of particles injected into a point. So, we see larger aerosols for point injections than for regional injections. To clarify, we change the sentence to: **In contrast to SO2 injections, point injections at 30N and 30S inject a higher number density of particles at the injection location, leading to more initial coagulation than regional injections (Benduhn et al., 2016). This results in a somewhat smaller effective radius for regional than for point injections for both MAM4 and CARMA (Figure 2c).**

Lines 206-208:  This is written as though it's surprising, but essentially these models are doing what they're designed to do.

Here, again, MAM4 is behaving very differently from CARMA. Later, we explain this by comparing the size distributions, and clearly, the behavior in MAM4 does not do what it is supposed to: the coarse model peak is not growing sufficiently in size.

Lines 222-223:  I found this sentence really frustrating.  It's written like a throwaway, but this is exactly the sort of thing that needs to be investigated further.

Based on the comment, we decided to add a little bit more explanation here. "**This indicates that differences in how the sigma range and size constraint of the modes in modal models are defined can lead to a significantly different aerosol burden for continuous sulfur injections**."

---

## Author Comment (AC2)

Response to Reviewers

We thank both reviewers for their helpful comments and suggestions, which we address in detail below. As a general remark upfront, we updated three figures in the revised version of the manuscript: Figure 2, panels a and b, and Figures 3 and 4. In Figure 2, we plotted the global burden in $TgSO_2$, but the title and figure caption showed TgS. Therefore, we correct the figure and now actually plot the burden in TgS. These changes do not affect the results or conclusions in the text. For Figures 3 and 4, we corrected a mistake in our plotting script for the modal aerosol model size distribution. The MAM4 size distribution, especially for H2SO4 injections (Figure 4), aligns better with the CARMA model. However, the same shortcomings in the nucleation mode for $SO_2$ injections and in the coarse mode in MAM4 persist. Finally, we changed the caption in Figure A5 from "Surface Temperature" to "Temperature" and added a zonal mean comparison of water vapor to Appendix A8, following the comment by Reviewer 1.

**Reviewer 1:**

Summary

This study systematically compares stratospheric aerosol intervention (SAI) simulations using two aerosol microphysical schemes – MAM4 (modal) and CARMA (sectional) – implemented within the same CESM2-WACCM6 framework, with all other model components identical. The experiments vary in injection material (SO2 vs. accumulation-mode H2SO4 aerosol), injection pattern (regional vs. point), and injection amount (5 vs. 25 TgS/yr). They show that the choice of microphysics alone can cause up to a twofold difference in simulated aerosol burden, and can even reverse the relative effectiveness of injection strategies (e.g., for $SO_2$, regional > point in MAM4 but the opposite in CARMA). These discrepancies arise because CARMA resolves a broader size distribution, leading to more nucleation of small particles and greater growth into coarse sizes, which enhances sedimentation and reduces total burden. They further show that these differences propagate to radiative forcing efficiency, stratospheric heating, and ozone responses, and that model divergence increases at higher injection rates.

Overall, this work provides a rigorous and timely evaluation of how aerosol microphysics shape SAI outcomes. As current SAI simulations (e.g., ARISE, GLENS) often rely on MAM schemes, this paper clearly points out that a better aerosol scheme might be needed when designing future community simulations. I recommend publication after minor revisions, as outlined below.

Comments

The SAI-induced stratosphere heating could also influence water vapor transport across the tropopause, which in turn changes radiative forcing as H2O is a GHG. I wonder if MAM4 and CARMA show different signals in stratospheric water vapor, and whether that could contribute to the difference in RF per injection (besides effects from aerosol burden and effective radius).

We thank the reviewer for this comment and added a new Figure in the appendix (Figure A8), which shows differences in water vapor. Clearly, the differences in stratospheric water vapor align with the amount of temperature increase in the lower tropical stratosphere for the different experiments. These changes are likely to cause a small positive radiative forcing that will counter some of the negative radiative forcing

from the aerosols, which is strongest for the regional injections and for MAM4 compared to CARMA. However, MAM4 radiative forcing for regional injections is still much stronger than for point injections, and also stronger compared to CARMA, suggesting a small effect from the water vapor changes.

To address this, we add the following test to the manuscript after discussion changes in lower tropical stratospheric heating: "**Differences in lower-stratospheric heating across the experiments also affect stratospheric water vapor, leading to a more substantial increase in water vapor with more heating for both CARMA and MAM4 (Figure A8). The increase in water vapor can partially offset the radiative forcing achieved by the sulfur injections.** "

Since community datasets like GLENS and ARISE use MAM, this work has direct implications for how we interpret those results. From Fig. 5b, the injection amount needed to reach a given forcing appears underestimated by roughly a factor of two in MAM4. Do the authors have any recommendations or comments on how ARISE or similar datasets should be used or interpreted moving forward?

As stated in the conclusion, there could be significant uncertainty in the SAI response to surface climate when using different aerosol models. One has to keep in mind that model results are always based on a specific model version and the experimental design, and one has to be cautious about a potential over- or underestimation of impacts. Indeed, required injection amounts in CARMA are much larger to reach the same RF and surface temperature response. This results in differences in different quantities. Here, we assessed two main quantities: stratospheric temperature change, which is important for changes in regional rainfall patterns, and effects on ozone. In general, we show that increases in stratospheric temperature between MAM4 and CARMA can differ by around 25%, with a bigger difference for larger injections. We also show that, dependent on the injection strategy (regional vs point injections), stratospheric temperatures can change. To address this comment, we modify this sentence in the conclusions to: "**The difference in the results highlights a significant uncertainty in the SAI response to surface climate, not only across different injection strategies but also across different aerosol models. To investigate the full effects of these differences on climate, future studies using an Earth System model with a fully coupled ocean are required.**"

Regarding ozone changes, we do show in part larger differences between MAM4 and CARMA, with the largest differences for SO2 injections, with 7% and 4-5% column ozone change per W/m2 in the SH polar regions for CARMA and MAM4, respectively. Analyzing process-based changes is more meaningful than quantifying climate impacts.

The study is based on single runs. I think this is fine since the focus is on comparing MAM4 and CARMA, but ozone and transport results (Section 6) could be somewhat sensitive to internal variability. A brief comment on this limitation would be helpful.

For this study, we performed 20 years for each injection experiment and used the last 10 model years for analysis. Since sea-surface temperatures are fixed, we don't think that the internal variability from the atmosphere alone will require additional years of simulation.

This paper involves multiple dimensions of comparison (SO2 vs. AM-H2SO4, regional vs. point, 5 vs. 25 TgS/yr), and at times these contrasts are discussed simultaneously, including some "difference in difference" discussions. Though the paper is generally very well written and I can infer meaning from context, many spots were confusing on the first read. I recommend adding clearer transitions when switching between

comparison dimensions and explicitly stating "compared to what" (though it may add some redundancy, it's helpful to improve clarity).

We agree with the reviewer and slightly revised the text here and at some places, and hope it is easier to follow in the revised manuscript.

Two examples below, but this recommendation applies to the manuscript as a whole:

Lines 132-133: "CARMA results in a larger … for point injections …". Is it "larger" than regional injection, or MAM4? Can also add reference to fig 2b here. To clarify, we added "than regional injections".

Line 190-199: These two paragraphs are particularly confusing on first read, because all comparisons are mixed and it's hard to track the logic. Maybe the authors could clearly state the structure, and use first paragraph to summarize results consistent across MAM4 vs. CARMA, and the second to highlight contrasting conclusions?

This paragraph introduces the similarities and differences between MAM4 and CARMA, which are discussed afterwards. We hope our modifications improve the sentence.

Technical corrections:

Line 280-281: explicitly introduce "SAD" in line 280 before using it in line 281.
Line 369: "TCO ozone" is repetitive
Line 373-374: parenthesis not closed

Thanks for the comments. We have corrected them.